# Recent Advances in the Application of Hydrogels as Drug Carriers in Inflammatory Bowel Disease: A Review

**DOI:** 10.3390/ijms26072894

**Published:** 2025-03-22

**Authors:** Qingrui Zhang, Bingxuan Lv, Manyu Li, Tiancai Zhang, Haoyu Li, Huimin Tian, Yanbo Yu

**Affiliations:** Shandong University Qilu Hospital, Jinan 250062, China; 202435761@mail.sdu.edu.cn (Q.Z.); lvbingxuan369@126.com (B.L.); limaryyy@163.com (M.L.); tiancaiya00@163.com (T.Z.); lihaoyu1232011@163.com (H.L.); 202435763@mail.sdu.edu.cn (H.T.)

**Keywords:** hydrogels, inflammatory bowel disease, colitis, drug carriers, drug delivery

## Abstract

Inflammatory bowel disease (IBD) is a chronic and refractory disease with increasing incidence, adversely impacting millions of patients worldwide. Current therapeutic strategies for IBD often exhibit considerable adverse effects, limited efficacy, and a high tendency for recurrence, highlighting the urgent need for novel therapeutic agents. Hydrogel, a three-dimensional hydrophilic network polymer material known for its excellent biocompatibility and responsiveness to stimuli, has been effectively utilized as a drug carrier across various therapeutic systems. The hydrogels’ application in IBD treatment holds significant promise for enhancing therapeutic outcomes. This review synthesizes recent advancements in leveraging hydrogels as drug carriers for IBD management. The discussion encompasses the response mechanisms of hydrogels, their application in IBD therapy, and methods of administration. As drug delivery matrices, hydrogels exhibit considerable potential for treating IBD.

## 1. Introduction

IBD, a persistent inflammatory condition encompassing Crohn’s disease (CD), ulcerative colitis (UC), and indeterminate colitis (IC), primarily manifests through symptoms, including weight loss, rectal bleeding, and diarrhea [1]. In the last decade, the incidence of both CD and UC has been steadily increasing, with projections indicating continued growth in incidence rates through 2050 [2,3]. While developed countries currently exhibit the highest incidence of IBD, the most rapid increases are observed in newly industrialized nations. For instance, in China, with its population of 1.4 billion, IBD incidence is anticipated to escalate in conjunction with accelerating urbanization and industrialization [4]. The resultant rise in healthcare expenditures related to IBD will impose significant burdens on patients, families, and society. Consequently, advancing IBD treatment modalities is of paramount importance.

Patients diagnosed with IBD primarily undergo medical management, utilizing aminosalicylates, glucocorticoids, immunomodulators, biologic therapies, and antibiotics as the cornerstone treatments. However, the therapeutic effects of these drugs are often unpredictable and inconsistent, frequently leading to recurrent disease episodes and the emergence of adverse reactions. For instance, therapeutic biologics induce immunosuppression through targeted immune modulation, conferring heightened susceptibility to opportunistic infections [5,6]. Additionally, the administration of olsalazine may result in diarrhea [7], whereas glucocorticoids can induce side effects, such as mood disorders, hypertension, hyperglycemia, osteoporosis, and/or increased susceptibility to infections [8]. Due to the inadequate targeting capabilities of conventional medications, patients often encounter the drawbacks of increased side effects, poor efficacy, and a propensity for the development of resistance to long term use [9]. Hence, it is requisite to develop a highly targeted drug carrier to improve the current situation of numerous side effects and poor targeting in IBD drug therapy. As a drug carrier, hydrogels have the capacity to respond to multiple stimuli and ultimately facilitate the precise release of drugs, making them an ideal carrier for IBD drug therapy.

Hydrogels are three-dimensional networks composed of hydrophilic polymers that have the ability to absorb and retain large amounts of water [10,11]. Due to their good biocompatibility and responsiveness to stimuli, hydrogels have been extensively utilized as a material for treating various biological systems [12], including rheumatoid arthritis [13], nervous system inflammation [14], periodontitis [15,16], uveitis [17,18,19], and increasingly, IBD. Hydrogels are sensitive to the environment of the digestive tract (such as pH and enzymes), can be released at specific sites, and possess certain targeting properties, thereby reducing the side effects of drugs. Additionally, hydrogels can prolong the action time of drugs at the inflammation site, better combat inflammation, and enhance the bioavailability and therapeutic effects of drugs. This review systematically introduces hydrogels’ response mechanisms, their applications in IBD treatment, and various hydrogel administration methods.

## 2. Response Mechanism of Hydrogels

PH-sensitive, temperature-sensitive, enzyme-sensitive, and ROS-sensitive hydrogels will be presented in the subsequent sections. Table 1 systematically summarizes the characteristic advantages and underlying mechanisms of hydrogels, categorized by their distinct response mechanisms.

### 2.1. pH-Sensitive

Commonly used pH-sensitive hydrogels include alginate (SA), hyaluronic acid (HA), chitosan (CS), and guar gum bean (GG) [27]. Hydrogels used for delivering intestinal drugs are predominantly anion-responsive, featuring acidic groups, such as carboxyl (-COOH) and hydroxyl (-OH) [12]. When pH increases, these acidic pendant groups dissociate hydrogen ions, leading to a large negative charge, hydrogel swelling, and rupture, releasing the loaded drug [28].

Curcumin (CUR), a type of natural polyphenol compound, has been shown to substantially alleviate symptoms in individuals with colitis [29]. However, the clinical utility of curcumin is constrained by its low bioavailability [30]. PH-sensitive slow-release hydrogels could serve as a potentially effective method for enhancing CUR delivery, representing a promising smart drug delivery platform. Zhang et al. designed a pH-sensitive hydrogel for loading CUR by preparing CC microspheres through the cross-linking of CS-derived carboxymethyl chitosan (CC) with phosphate. The HA-GE hydrogel was produced by chemically crosslinking HA with gelatin (GE). Subsequently, CC microspheres loaded with CUR were embedded within the HA-GE hydrogel matrix. In HA-Ge hydrogels, a network structure comprising HA and Ge is established through the cross-linking of amide bonds. During preparation, the swelling behavior of the HA-Ge hydrogel is influenced by the HA to GE ratio. With higher GE concentrations, increased cross-linking with HA results in a tighter network and a lower swelling rate. Conversely, as HA concentration increases, the maximum swelling rate rises. Within an alkaline environment, the polyanionic nature facilitates drug release from the hydrogel via swelling and degradation of the hydrogel network, thereby enabling colon-specific release and sustained CUR administration during IBD treatment. Hydrogels made of HA-GE show minimal expansion in conditions mimicking stomach acid with a pH of 1.2. In conditions when pH = 6.8/7.4, resembling intestinal fluid, there is a noticeable expansion and eventual rupture, leading to the release of CUR. Under the protection of hydrogel, the release time of CUR was prolonged, and the therapeutic effect was significantly enhanced at the colon site [Figure 1A] [20].

Gallic acid (GA) is a compound found in nature. GA can suppress the NF-κB signaling and elevate the levels of inflammation-related cytokines [31]. The clinical application of GA is restricted because it is quickly eliminated, has a short retention period in the digestive system, and is poorly absorbed when taken orally [32]. Huang et al. developed a hydrogel composed of gallic acid and sodium alginate (GAS), featuring resistance to gastric acid, prolonged gradual release, and properties that adhere to mucosal surfaces. The stability of GAS hydrogels was assessed in various gastrointestinal fluids at 37 °C. In artificial colon fluid, the degradation rate was rapid, reaching 70% within 12 h. In contrast, in artificial gastric fluid, the degradation was considerably slower, indicating enhanced stability in acidic conditions. This acid resistance may result from ionic interactions between hydrogen ions and deprotonated sodium alginate, which help maintain hydrogel stability at lower PH. Conversely, higher pH values disrupt these ionic interactions, increasing degradation [21]. The incorporation of drugs into pH-sensitive hydrogels can substantially enhance drug bioavailability.

### 2.2. Temperature-Sensitive

Thermosensitive hydrogels undergo sol-gel-sol transitions as temperature changes. Based on this feature, several temperature-sensitive hydrogels for rectal administration have recently emerged. At room temperature, they are freely flowing sols. Upon exposure to body temperature, they naturally transform into semi-solid hydrogels, exhibiting excellent injectability and rapid, uncomplicated gelation properties, thus facilitating enema administration [Figure 1B] [33,34].

The poly (aliphatic ester)-based triblock copolymer, poly (D, L-lactic acid)–poly (ethylene glycol)–poly (D, L-lactic acid) (PDLLA–PEG–PDLLA), remains in a solution state at room temperature but transitions to a gel state at body temperature when the polymer concentration exceeds 11%. The copolymer PDLLA-PEG-PDLLA is characterized by its favorable biosafety profile, affordability, and straightforward synthesis process. Guo et al. utilized modulus changes to assess the viscosity of P4 hydrogel under varying stabilization conditions. At temperatures below room temperature (25 °C), P4 solution exhibited a very low modulus, indicating a free-flowing sol state and excellent injectability. As the temperature increased, a sharp rise in modulus was noted, accompanied by the formation of a semi-solid hydrogel. The modulus peaked at 37 °C, suggesting effective injectability in vivo. They mixed PDLLA-PEG-PDLLA and mesalazine to prepare a kind of hydrogel loaded with mesalazine (Mesa/Gel). Following the administration of DSS mice enema, the solution quickly converted into gels on the colon’s surface due to the influence of body heat. These gels served as a physical barrier, effectively preventing bacterial invasion. Mesa/Gel can also enhance drug retention and release in the colon by providing strong adhesion [22].

Xu et al. [23] designed a biocompatible and IBD-targeted metabolic nanoregulator (TMNR). The temperature-sensitive hydrogel FCH was first formed by cross-linking Pluronic F127 (F127), α-cyclodextrin (α-CD), and HA. At room temperature (4 °C), FCH was in a liquid state, while at body temperature (37 °C), it transformed into an opaque white gel. Upon the addition of 3 wt% α-cyclodextrin (α-CD), the FCH hydrogel exhibited a threefold increase in stability compared to the F127-HA hydrogel, maintaining resistance to degradation for up to six days. Rheological properties were assessed using strain oscillatory shear experiments, revealing that the viscosity of FCH increased with temperature. The elasticity of FCH was confirmed through modulus measurements. These findings indicate that FCH hydrogel is well-suited for the intestinal temperature and peristaltic environment. A melanin-gallium complex (MNR) was added to FCH to form an injectable TMNR. To evaluate the adhesion capacity of TMNR within intestinal tissues, Rhodamine B (RhB)-stained TMNR was administered to Sprague-Dawley (SD) rats via enema. The findings indicated that the pink colloidal hydrogel demonstrated robust wet adhesion to the intestinal surface. The TMNR maintained exceptional resistance to displacement even when the intestinal segment was inverted and twisted. Additionally, the continuous injection of phosphate-buffered saline (PBS) solution into the hydrogel did not affect its position or adhesion strength. These results underscore the hydrogel’s substantial adhesive capability to intestinal lesions amidst gastrointestinal peristalsis. TMNR was administered by enema to healthy and DSS mice. TMNR is able to improve colitis-targeting ability and extend the duration of colitis therapy retention. The results showed that this thermosensitive hydrogel has excellent colitis targeting ability, prolonged residence time, adhesive ability, and physical barrier function [23].

### 2.3. Enzyme Sensitivity

Pectin (PC) remains intact in the acidic conditions of the stomach and small intestine but can be broken down by enzymes in the colon [35,36]. Hydrogels responsive to pH and enzyme activation using PC and pH-sensitive polyacrylamide (PAM) for targeted delivery of budesonide to the colon have been developed [37]. These hydrogels demonstrate swelling behavior influenced by pH levels and a two-phase drug release pattern.

Quercetin (QT) is a naturally occurring flavonoid found in the diet, known for its safety profile and a range of pharmacological activities that combat inflammation. However, oral QT is not effective for treating IBD because it has low solubility and undergoes significant metabolism in the digestive system [38]. Jing et al. [24] developed a QT delivery system aimed at the colon (COS-CaP-QT); they prepared and crosslinked the pectin/Ca^2+^ microspheres by oligo-chitosan (COS). In comparison to high molecular weight chitosan, low molecular weight chitosan offers advantages, such as reduced steric hindrance, lower diffusion resistance, and enhanced penetration capabilities. The coating formed by high molecular weight chitosan on microspheres is typically thinner, whereas the coating from low molecular weight chitosan undergoes greater solidification during liquefaction and demonstrates robust anti-swelling properties. This indicates that low molecular weight chitosan, COS, has superior film-forming properties. Oligosaccharides enhance the stability of CaP-QT microspheres more effectively than high molecular weight chitosan, likely due to their ability to not only cover the surface but also deeply penetrate the microspheres. COS can also undergo degradation via enzymes and bacteria specific to the colon [38]. To investigate the influence of the colonic environment on COS-Ca^2+^P-QT particles, rats were administered COS-CaP microspheres devoid of the QT drug. Subsequently, colonic mucosa was harvested and homogenized in a PBS solution to produce colonic mucosal homogenate (CMH). The COS-Ca^2+^P-QT particles were mixed with a phosphate solution, CMH, and its inactive form (i-CMH). After being combined with CMH, the particles quickly broke apart, yet they remained mostly intact in PBS and i-CMH. This suggests that COS-CaP-QT microspheres are sensitive to the colon-specific microenvironment and have drug protective and colon-targeting effects [24].

HA is susceptible to degradation by ROS and hyaluronidase in the intestines [39] and therefore targets the inflamed colon. Recently, Huai et al. prepared an oral hydrogel (AL/HA hydrogel) designed to respond to the colon’s microenvironment for the treatment of IBD. This was achieved by linking natural sodium alginate (AL) and HA mediated by Ca^2+^/Zn^2+^ under mild conditions. To investigate the release characteristics of HA/AL hydrogels loaded with drugs in the gastrointestinal tract in a more logical manner, they evaluated the in vitro drug release profiles of the hydrogels by modeling two drugs, FITC-labeled BSA and IgG, and simulating the gastrointestinal environment [Figure 1C]. In simulated gastric juice environment, the release rates of IgG_FITC_ and BSA_FITC_ were merely 3.2% and 5%, respectively. Under conditions mimicking the small intestine, the sustained release of BSA_FITC_ and IgG_FITC_ was achieved at 3 h with release rates of approximately 30% and 23.4%, respectively, after which the release rate slowed significantly. In a simulated enteritis environment, AL/HA@BSA_FITC_ and AL/HA@IgG_FITC_ hydrogels exhibited an increased release, achieving rates of 60% and 52.1%, respectively. Following this period, the primary trend of drug release slowed down. These results indicated that the AL/HA hydrogel has promising characteristics for responding to the colonic microenvironment and holds significant biological importance, especially in the context of colitis [25].

### 2.4. ROS Sensitive

The generation of ROS facilitates the oxidation of sulfhydryl groups into disulfide bonds. Consequently, sulfhydryl-substituted polymers can specifically form hydrogels at inflammatory sites [40]. Building upon this redox-responsive mechanism, Zhang et al. engineered thiol-functionalized HA into a tunable hydrogel system. Controlled thiolation ratios (20–60%) generated HASH variants, with modification efficiency quantified via NMR deconvolution and FTIR sulfhydryl peak analysis. Gelation kinetics under varying H_2_O_2_ concentrations revealed concentration-dependent depletion of free thiol groups, with HASH_60%_ exhibiting the fastest crosslinking rate, thereby validating ROS-triggered HASA hydrogel formation. In vitro colonic tissue assays validated HASA’s intestinal adhesion, where H_2_O_2_ did not affect the adhesion of HA while improving the bioadhesion of thiolated derivatives (HASA_60%_ demonstrating superior adhesion). This enhancement mechanistically stems from covalent disulfide bridging between HASH-derived thiols and gut surface thiols via dynamic thiol-disulfide exchange [41]. CY5.5-labeled HASA and HA were administered intraorally to assess in vivo biodistribution in healthy versus DSS-induced colitis mice. HASH_60%_-treated colitic mice exhibited pronounced colorectal fluorescence at 8 h post-administration, with inflamed colons demonstrating higher colonic adhesion compared to healthy counterparts, indicative of ROS-mediated oxidation into hydrogels specifically at inflammatory loci. However, no hydrogel was detected in the sections obtained from healthy mice, which underscores the targeted adhesive properties of HASH specifically to the inflamed intestinal tissue [26].

## 3. Application of Hydrogels as Drug Carriers in the Treatment of IBD

Drug-loaded hydrogel systems exert anti-inflammatory effects in intestinal pathologies through multiple mechanistic pathways. Table 2 comprehensively summarizes their therapeutic efficacy and corresponding experimental models.

### 3.1. Maintained the Intestinal Barrier

The intestinal mucosal barrier, primarily composed of tight junctions between the mucous layer and epithelial cells, is crucial for nutrient absorption and pathogen exclusion [54]. Damage to the intestinal mucosal barrier is a key pathological feature of IBD. Hydrogels can act as physical barriers to prevent pathogen invasion in IBD therapy.

Guo et al. developed a thermosensitive hydrogel capable of gelling in the intestines to treat DSS mice via enema. Even without drug loading, hydrogel injection significantly alleviated symptoms in mice with colitis. This phenomenon is primarily attributed to the hydrogel’s physical blocking effect, which effectively hinders the penetration of microorganisms within the intestinal lumen. The hydrogels present in the colon shield the colon cells from exposure to destructive elements. Additionally, the gel’s macromolecular structure prevents rapid uptake, thus preserving endogenous growth factors that aid in regeneration and allowing colon cells to self-repair. In an in vitro study, *Staphylococcus aureus* and *Escherichia coli* were cultured in the upper chamber, with the hydrogel positioned between the upper and lower chambers. The hydrogel’s ability to block bacterial penetration was assessed by measuring the bacterial optical density (OD) in the lower chamber after 18 h. Following 18 h of incubation, the optical density of bacteria in the lower chamber of the gel group was notably less than that in the upper chamber. The gel effectively obstructed bacterial movement, serving as a physical barrier [22].

Inulin is a naturally occurring linear fructan extracted from various edible plants. As a prebiotic fiber, it can be enzymatically degraded by inulinase in the colon, demonstrating high biocompatibility and safety. The abundant hydroxyl groups present in inulin facilitate strong cross-linking and hydrogel formation, enhancing bioadhesion and colon retention. Consequently, inulin is chosen as a polysaccharide for hydrogel construction. Recently, scientists developed an omeprazole-derived nanoneedle and inulin gel [Cu_2_(Olsa)/gel] and evaluated the therapeutic effect by evaluating intestinal tissue changes in rats affected by colitis induced by DSS. The DSS-treated group, which did not receive any additional intervention, exhibited disruption in the crypt architecture and a notable decrease in goblet cell count. In contrast, the Cu_2_(Olsa)/gel group demonstrated goblet cell numbers that were comparable to those of the normal group, with a well-maintained structure of the intestinal tissue. The expression of occludin-1 (OC-1) and ZO-1, which are key proteins that play a crucial role in the formation of tight junctions within intestinal epithelial cells and influence the permeability of the intestinal barrier [55], was increased in the Cu_2_(Olsa)/gel-treated group. Simultaneously, the colonic epithelium and microvilli in the Cu_2_(Olsa)/gel treatment group remained relatively more intact. Treatment of TNBS-induced colitis in mice, followed by histological analysis, revealed that Cu_2_(Olsa)/gel significantly attenuated neutrophil infiltration and epithelial damage in colonic tissue. Additionally, Cu_2_(Olsa)/gel facilitated tissue repair and mucus secretion, thereby restoring intestinal barrier function. The experimental findings indicated that the application of Cu_2_(Olsa)/gel significantly improved the function of the intestinal epithelial barrier, a crucial factor in preventing the movement of bacteria and reducing inflammation [42].

Yang et al. [43] developed a Mn_3_O_4_ nanozyme-loaded thermosensitive hydrogel (MLPPP) using the thermosensitive hydrogel PDLLA-PEG-PDLLA for the treatment of DSS-induced colitis. In vitro experiments demonstrated that MLPPP exhibited a degradation rate of approximately 87% at 72 h, indicating that the core drug, Mn_3_O_4_, can rapidly and effectively target sites of intestinal alteration to inhibit colitis progression. In mice with colitis, there was an observed increase in necrotic cells, severe mucosal damage, heightened inflammatory cell infiltration, and destruction of crypt structure. Conversely, colitis-afflicted mice treated with MLPPP exhibited an almost normal histological microstructure, closely resembling that of healthy mice. Goblet cells secreting MUC-2 mucin protect the colon, and Paneth cells may reduce the risk of colitis. The MLPPP-treated group showed increased levels of MUC-2 expression, a significant increase in goblet cell counts, and demonstrated protection against DSS-induced apoptosis in the colon in contrast to the control group. The expression levels of CLDN1, OCLN, and ZO-1 were significantly elevated in MLPPP-treated mice compared to those in the DSS group. The restoration of the intestinal lining’s protective layer is beneficial, underscoring the protective role of MLPPP on mucosal epithelial defense [43].

A fully synthetic peptide solution known as self-assembling peptide hydrogel (SAPH) offers an alternative to collagen, consisting of 16 amino acid groups with repeated sequences of alanine, aspartic acid, and arginine. Upon exposure to physiological pH or tissue, the number of beta-sheet structures between SAPH monomers increases, leading to gel thickening and the formation of nanofiber hydrogels. These hydrogels mimic natural extracellular matrix scaffolds, thereby facilitating cell and tissue preservation during the healing process [56]. Araki et al. demonstrated, through animal experiments, that local application of SAPH can enhance the healing of colonic ulcers. Initially, TNBS was employed to induce colitis in rats, followed by the topical application of SAPH to the ulcers via endoscopic treatment. In the SAPH-treated group, there was a significant reduction in ulcer area accompanied by extensive regeneration of epithelial cells around the ulcer base, indicating that local SAPH application can expedite the ulcer recovery process. Subsequently, endoscopic procedures were employed to create mechanical wounds in the distal colon of the rats. Endoscopic examination revealed that the model treated with SAPH showed a significantly faster reduction in wound diameter and area compared to the vehicle (normal saline solution). The application of SAPH also increased the expression of factors related to wound healing, such as Cldn1 and Vil1 [44].

### 3.2. Regulate the Immune System

Indicators for detecting immune responses in IBD primarily include macrophage differentiation, cytokine generation (promoting or reducing inflammation), and inflammatory signaling pathways, such as MAPK and NF-κB. Liu et al. developed and fabricated adhesive core-shell hydrogel microspheres (HAMs) targeted to the colon. Uniformly sized HA-SH-Ag hydrogel microspheres (HMs) were created using a gas shearing method. Calcium diffused from the interior of the microspheres, cross-linking to the alginate surface, ultimately forming HAMs. M1 macrophages are generally associated with inflammation, tissue damage, and inhibition of wound healing [57], while M2 macrophages counteract inflammation and facilitate tissue repair [58,59]. In vitro, HAMs significantly suppressed MAPK and NF-κB signaling pathways, antagonized inflammation, and induced M2 macrophage polarization. Animal experiments using immunofluorescence analysis of colon tissue macrophages revealed that HAMs promoted M2 macrophage differentiation while suppressing M1 differentiation in DSS-induced colitis. HAMs also reduced pro-inflammatory cytokine expression and secretion and increased the cytokine transforming growth factor-β (TGF-β), associated with tissue repair [45].

Selenoproteins are crucial for regulating immune cells and controlling inflammation, but their oral bioavailability is poor due to facile degradation by gastric acid [60,61]. Scientists synthesized hydrogel microbeads (SHSe) by encapsulating Se nanoparticles modified with hyaluronic acid (HA-Se) within a calcium alginate (SA) hydrogel shell. Intestinal macrophages from IBD patients may produce high levels of CD14, activate the NF-κB pathway, and release pro-inflammatory molecules [62]. Activated macrophages produce TNF-α, influencing stromal cell transformation into myofibroblasts and inducing epithelial cell apoptosis [63]. In vitro, HA-Se effectively inhibited the production of pro-inflammatory cytokines (TNF-α, IL-1β, and IL-6), cleared intracellular ROS, and inhibited the NF-κB pathway. Animal studies showed that SHSe microspheres mitigated intestinal inflammation in mice with IBD by decreasing neutrophil and monocyte levels, suppressing inflammatory cytokine release, and enhancing immune Treg cell levels [46]. These results demonstrate the excellent anti-inflammatory activity and maintenance of intestinal immune homeostasis by SHSe microbeads.

The severity of inflammation is indicated by symptom factors. A hydrogel encapsulated with emodin and curcumin (CUR/EMO NE@SA) has shown promise in reducing inflammatory factors in colitis mice. In colitis mice, plasma levels of pro-inflammatory cytokines IL-6 and TNF-α were significantly higher than in healthy mice. However, treatment with CUR/EMO NE@SA resulted in plasma and colon TNF-α and IL-6 levels similar to those in healthy mice, indicating no significant difference. CUR/EMO NE@SA effectively reduces inflammation by modulating immune responses, enhancing protective factors, and decreasing harmful cytokines [47].

### 3.3. Regulate Gut Microbiota

Dysbacteriosis involves changes in the gut microbiota’s structure and function, playing a crucial role in the development of IBD [64,65]. Stem cells obtained from umbilical cord blood (UCB-MSCs) show promise for regenerating damaged and diseased tissues in IBD. However, oral administration of UCB-MSCs faces challenges due to breakdown by stomach acid and digestive enzymes, resulting in low bioavailability. Encapsulation of UCB-MSCs within a hydrogel prior to oral administration significantly enhances their therapeutic efficacy. Kim et al. have recently introduced a hydrogel microcapsule containing stem cells, featuring a thin oil coating, which not only ensures the integrity of UCB-MSCs before they reach the inflamed site but also allows the shell to be broken by segmentation during intestinal motility and peristaltic processes. Thus, UCB-MSCs are delivered to the vicinity of inflamed colonic tissue and microbiota. The Chao1 index, a measure of organism richness within a given taxon, significantly increased in colitis mice treated with SC-HM, indicating effective correction of gut flora abundance imbalance induced by DSS. Bacteroidetes, abundant in healthy individuals and decreased in IBD patients, have anti-inflammatory effects [66]. SC-HM can also regulate Firmicutes and Bacteroidetes imbalance and improve intestinal mucosal immunity. Short-chain fatty acids (SCFAs) are crucial for intestinal cell energy and maintaining intestinal immune balance [67]. SC-HM may enhance the population of bacteria producing SCFAs. Oral administration of SC-HM assists in rectifying the gut microbiome imbalance and supports the restoration of gut microbiota composition during active colitis [48].

Zein/alginate core-shell microspheres loaded with bioactive glass (BG), denoted as Zein/SA/BG, were developed to treat DSS-induced colitis in mice. The impact of Zein/SA/BG on the gut microbiota in colitis-afflicted mice was investigated using 16S rRNA sequencing. DSS treatment significantly reduced gut microbiota richness. However, Zein/SA/BG treatment partially restored the relative abundance of bacterial flora. The S24-7 group associated with carbohydrate breakdown and gut lining restoration [68], showed a marked increase in abundance after Zein/SA/BG treatment. The family Peptostreptococcaceae, linked to various pathways and host genes in IBD [69], increased in inflamed intestines but decreased after treatment. Loss of biodiversity can disrupt intestinal barrier integrity, impair immune system performance, and potentially exacerbate inflammation and immune reactions [70]. Persistent inflammation decreased gut microorganism diversity, but Zein/SA/BG partially recovered this diversity. Zein/SA/BG may restore microbiota equilibrium by promoting the growth of microbiota associated with gut repair while inhibiting microbiota implicated in IBD pathogenesis [49].

### 3.4. Clear Intracellular ROS

Oxidative stress damages the mucosal barrier, facilitates pathogen entry, and activates inflammation in IBD, contributing to disease initiation and progression. ROS in the intestinal lining of IBD patients is strongly associated with immune cell accumulation and activation at the inflammation site [71]. The combined effect of ROS and inflammatory cells causes cellular damage and oxidative stress, worsening the inflammatory microenvironment and intensifying colitis symptoms [72]. Insulin-like growth factor 1 (IGF-1C) was conjugated to a chitosan (CS) hydrogel to form a CS-IGF-1C hydrogel. This hydrogel was injected orthotopically along with human placenta-derived mesenchymal stem cells (hP-MSCs) to treat TNBS-induced colitis in mice. Molecular imaging techniques were used to visualize ROS in real-time and track transplanted hP-MSCs using bioluminescence imaging. ROS levels were considerably elevated in the PBS group compared to the sham group. Conversely, ROS levels gradually decreased in the MSCs, MSCs/CS, and MSCs/CS-IGF-1C groups. Notably, the MSCs/CS-IGF-1C group achieved ROS levels nearly equivalent to those of the sham group. The combination of hP-MSCs with CS-IGF-1C hydrogel significantly decreased ROS levels, alleviating colitis symptoms in TNBS mice [50].

Sterically hindered amines (SHAs) rapidly convert to nitro radicals in oxidative environments, neutralizing ROS and various free radicals [73]. SHAs regenerate themselves and consistently neutralize free radicals via the Denisov cycle, enhancing their free radical scavenging capacity. A novel oral hydrogel formulation featuring sterically hindered amine-based redox polymers (SHARPs) chemically linked to antioxidant SHA entities was developed. The ROS scavenging ability of SHARP was tested in three independent models: peroxyl, hydroxyl, and superoxide radical models. Fluorescence probes susceptible to oxidation by free radicals were used to assess antioxidant scavenging properties. Without the polymer, the fluorescence quickly diminished in under 40 min, but SHARP hydrogel significantly counteracted this reduction. A link was observed between remaining fluorescence and SHARP occurrence, indicating the polymer’s successful blockage of peroxyl radical oxidative function. In the superoxide anion model, superoxide radicals convert nitro blue tetrazolium (NBT) into its methylene variant, increasing absorbance at 572 nm. UV-Vis spectrophotometric analysis showed approximately 40% scavenging activity at the lowest concentration (1.7 mg/mL) and 95% at a 20-fold concentration, confirming SHARP hydrogel’s high efficiency in inactivating superoxide anion radicals [51].

### 3.5. Adhesion

The colon’s mucosa allows for sustained drug release and enhanced effectiveness when using a drug loading system with mucosal adhesion [74]. HA was labeled with fluorescein (FL) to visualize hydrogel adhesion to the intestinal wall in both colitis and normal rats. In healthy colon tissue, a clear boundary existed between mucosa, chorionic villi, and blood vessels, with minimal macrophage presence in the lamina propria. In the TNBS-induced colitis group, damage was observed in both the chorionic villi and the mucosal layer, with the submucosal layer showing increased thickness compared to the standard control group. The FL-labeled HA conjugate bound to the mucosal surface and interacted with blood vessels. The HA-FL conjugate showed greater affinity for ulcerative mucosa compared to healthy colon [52].

### 3.6. Anti-Fibrosis

Chronic colitis often involves fibrosis, characterized by extracellular matrix component accumulation (e.g., collagen) in the intestines, leading to tissue rigidity and potential scarring [75]. Various cells contribute to fibrosis, including intestinal subepithelial myofibroblasts expressing α-SMA and smooth muscle cells [76]. A microsphere with a zein and sodium alginate core-shell structure incorporating bioactive glass (BG) was created to assess their impact on fibrosis in DSS-induced chronic colitis mice. α-SMA immunostaining was used for analysis. Saline and microsphere groups exhibited thicker α-SMA-expressing tissue layers compared to healthy mice. Within the microsphere group, a decrease in α-SMA-expressing tissue layer thickness was observed, suggesting a role for microspheres in reducing fibrosis-related cell types. Masson staining assessed collagen content. The saline group showed higher collagen levels in the muscle layer and submucosa compared to the microsphere group [49].

### 3.7. Relieve Anxiety and Depression

Clinical populations with IBD exhibit a statistically significant elevation in comorbid anxiety and depressive disorders, a phenomenon potentially mediated through dysregulation of the microbiota-gut-brain axis [77]. Gut microbial dysbiosis is pathologically characterized by the expansion of pro-inflammatory pathobionts that liberate proinflammatory mediators capable of translocating across the structurally and functionally compromised intestinal epithelium into systemic circulation [78]. Proinflammatory mediators compromise blood-brain barrier (BBB) integrity, inducing neuroinflammatory cascades and neurodegeneration that predispose to neuropsychiatric pathologies [79]. Rhein (Rh) has anti-inflammatory, antioxidant, anti-tumor, and neuroprotective effects [80]. Spirulina platensis (SP) is a natural microalga that has been approved by the FDA as a functional food [81]. SP demonstrates gastric-resistant structural stability during gastrointestinal transit, with bioadhesive properties enabling selective retention within intestinal villi for optimized luminal distribution and superior oral bioavailability. Zhong et al. developed a dual-functional hydrogel system that ameliorated both colitic manifestations and neurobehavioral comorbidities in chronic colitis mouse models. The rhein-based hydrogel (Rh-gel) was fabricated via supramolecular self-assembly, followed by spirulina SP incorporation to engineer a hierarchical 3D-structured composite (SP@Rh-gel). Behavioral assessments, including open field (OFT), elevated plus maze (EPM), forced swimming (FST), and tail suspension (TST) tests, were conducted in colitis mouse models post-treatment. S100 calc-binding protein β subunit (S100β) is a biomarker of brain injury and BBB destruction [82]. Results demonstrated that DSS-induced colitis elicited neuropsychiatric comorbidities, whereas SP@Rh-gel administration significantly attenuated anxiety- and depression-like phenotypes through gut-brain axis modulation. DSS-induced chronic colitis elicits neurotoxicity via microglial NLRP3 inflammasome activation, driving hippocampal neuroinflammation and impaired neuroplasticity. SP@Rh-gel attenuated microglial reactivity, suppressing hippocampal caspase-1/NLRP3/IL-1β expression and normalizing S100β levels, indicating BBB restoration. Mechanistically, SP@Rh-gel preserved intestinal barrier integrity while reducing systemic proinflammatory cytokine translocation to the hippocampus. Gut microbiota analysis revealed therapeutic modulation of inflammation- and depression-associated taxa through metabolic reprogramming. This hydrogel offers translational potential for IBD-related neuropsychiatric comorbidities [53].

## 4. Mode of Administration

Hydrogels have demonstrated distinctive advantages as drug delivery vehicles across various administration routes. Table 3 provides the benefits associated with each administration pathway.

### 4.1. Oral Administration

Oral administration is the most common IBD treatment method due to its simplicity and high patient compliance. However, drug absorption or breakdown in the upper digestive tract is a significant limitation. Hydrogels as drug carriers can overcome this by preventing premature drug release and degradation. The most prevalent hydrogel carriers employed for oral administration are CS and SA.

Chitosan, a biopolymer derived from chitin through deacetylation, comprises β-(1-4)-linked D-glucosamine and N-acetyl-D-glucosamine [88]. Chitosan’s branched chains contain numerous amino groups, making it pH-sensitive. This is also the basis for CS protection against gastric acid destruction. CS exhibits good biocompatibility [89,90] and is susceptible to human enzymatic hydrolysis. Alginate is a linear polysaccharide copolymer composed of _D_-mannuronic acid and L-guluronic acid units, linked together by 1,4-glycosidic bonds [91]. SA contains a large number of carboxyl groups [92]. At low pH, the protonation of the carboxyl group leads to the contraction of SA hydrogels. In high pH environments, such as the intestine, SA generates a large amount of mutually exclusive charge, causing the hydrogel to swell. Qiu et al. encapsulated the drug with alginate saline gel microspheres to address the challenges posed by stomach acidity and digestive enzymes [83]. Hydrogel microspheres play a crucial role in safeguarding the medication from harsh acidic conditions and various enzymes, ensuring its delivery to the colon and rectum. Liu et al. loaded oral drugs in chitosan/alginate saline gels, which are pH-sensitive and swell and rupture in the colon’s alkaline conditions to deliver the medication [84]. Hydrogels serve as carriers for oral medications, shielding them from breakdown caused by stomach acid and digestive enzymes, which enhances the drugs’ stability and efficiency.

### 4.2. Rectal Administration

Rectal administration, involving injection into the rectum via syringe or enema, can bypass first-pass metabolism and act directly on the primary lesion site, mitigating side effects and enhancing bioavailability [93,94]. However, rectally administered drugs are often liquid, which results in high mobility and brief contact with the intestinal mucosa, thereby limiting their effectiveness. Moreover, patients with active distal colitis typically cannot tolerate liquid enemas [95]. Incorporating drugs into hydrogels may extend their duration of action within the intestinal tract and improve their therapeutic effectiveness.

Scientists have devised a temperature-sensitive drug-loaded hydrogel for rectal administration. This hydrogel is a liquid at room temperature, facilitating injection, and can transform into a gel state at body temperature. The material demonstrates strong adhesion to biological tissues and facilitates quick medication release, aiming to achieve a swift impact on the compromised intestinal lining [96]. Hydrogels impart heat sensitivity and adhesiveness to drug delivery systems, enhancing their delivery, scalability within the mucosal layer, and persistence in the rectal passage. Zhang et al. selected ascorbyl palmitate (AP)—a hydrogel degradable by enzymes from inflammatory cells—as a carrier for dexamethasone in the formulation of an IT-hydrogel enema. Characteristics of inflamed mucosa include reduced mucus, increased positively charged proteins, and heightened epithelial cell permeability. The IT-hydrogel with a negative charge fails to stick to normal intestinal walls but does adhere to positively charged inflamed intestinal walls. Inflammatory cells release hydrolytic enzymes that break down the gel, releasing the drug. The study demonstrated that negatively charged hydrogel fibers preferentially bound to positively charged artificial surfaces, inflamed tissues in mouse models of colitis, and biopsy specimens from inflamed colonic tissue of human patients with UC. In addition, IT-hydrogel has good biocompatibility, a low price, easy preparation, and a strong drug loading capacity. The IT-hydrogel, impregnated with medication, demonstrates prolonged stability, enabling a sustained release of the drug over several days. This characteristic reduces the frequency of administration, lessening the patient’s burden and enhancing treatment adherence [85].

### 4.3. Endoscopic Administration

Endoscopic surveillance constitutes the principal diagnostic approach for IBD. Administering therapy during endoscopy can greatly enhance the accuracy of the delivery site, thereby integrating diagnosis and treatment. Yoon et al. designed a hydrogel capable of achieving a temperature-responsive solution-gel conversion and sprayed it endoscopically onto inflamed bowel walls to integrate examination and treatment. They conjugated the hydrogel with peptides that serve as contact signal transduction initiators, endowing the hydrogel with lesion-specific adhesion and enhancing therapeutic efficacy by minimizing adverse effects on non-lesion-affected areas. Hydrogel adhesion can physically improve barrier function, restoring cell-cell interactions and mitigating inflammatory injury. The gel’s characteristics can significantly enhance the duration of contact between the drug and the intestinal wall, thereby extending the therapeutic effect. This delivery method minimizes drug exposure to healthy tissue and significantly reduces side effects [86]. Their experimental results offer novel ideas for the all-in-one treatment of IBD.

### 4.4. Subcutaneous Administration

Currently, the use of drug-containing hydrogels for subcutaneous injection in IBD management is limited. However, some studies have discovered that administering drug-loaded hydrogels subcutaneously can promote the repair of hard and soft tissues in mice with periodontal tissue injury [97]. Prolyl hydroxylase (PHD) inhibitors, specifically DPCA, can enhance HIF-1α levels and promote the regeneration of mammalian cells [98]. DeFrates and colleagues conjugated DPCA with polyethylene glycol (PEG) to form a self-assembling hydrogel, which they injected subcutaneously into the posterior neck region of DSS-induced mice. Although the subcutaneous injection did not directly contact the colon, they observed up-regulation of HIF-1α in the mouse colon. This supramolecular hydrogel can gradually release DPCA to achieve stable release of DPCA. In vivo results showed that a single subcutaneous injection of PEG-DPCA could completely restore the healthy tissue structure of mice with colitis [87]. Subcutaneous injection of drugs has a long time of action, slow release, and simple administration process, which can enhance patient adherence. Introducing subcutaneous injection into IBD treatment holds considerable importance.

## 5. Conclusions and Prospect

Incorporating hydrogels as drug delivery vehicles for IBD management is a promising strategy, given the numerous side effects, poor targeting, and drug resistance associated with current therapies. Hydrogels exhibit remarkable responsiveness to various stimuli, enhancing drug delivery targeting. This capability minimizes drug-induced irritation of healthy intestinal tissue, reduces adverse effects, and improves drug stability. Hydrogels are customizable, addressing IBD’s pathological characteristics and enabling concurrent treatment of multiple therapeutic targets [Figure 2]. They demonstrate excellent biocompatibility and natural breakdown. Hydrogels can be configured for diverse drug delivery modalities, supporting personalized treatment regimens.

However, hydrogel-based drug delivery systems face challenges. Ensuring non-toxicity and non-immunogenicity in humans requires rigorous animal studies and preclinical trials. In vivo experiments have limitations in observing the complete therapeutic process, necessitating real-time, monitorable indicators for comprehensive disease assessment. For chronic IBD requiring prolonged therapy, it is crucial to prolong drug effect, decelerate release, and define dosing schedules to maintain therapeutic blood levels. Discontinuous, heterogeneous, and widespread IBD lesions pose a challenge in ensuring comprehensive drug contact with all affected areas. Hydrogels hold significant potential for IBD treatment, and ongoing research will undoubtedly lead to superior therapeutic options.

## Figures and Tables

**Figure 1 ijms-26-02894-f001:**
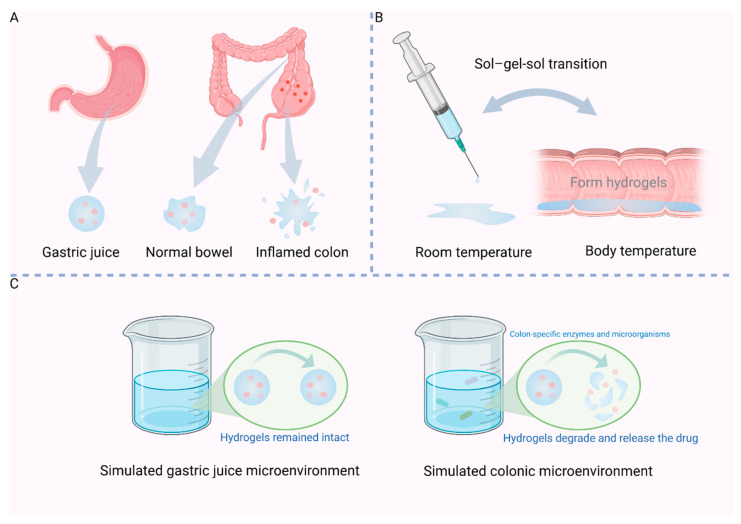
Response mechanism of hydrogels. (**A**) pH-sensitive hydrogels. (**B**) Temperature-sensitive hydrogels. (**C**) Enzyme-sensitive hydrogels. Created by Biorender.

**Figure 2 ijms-26-02894-f002:**
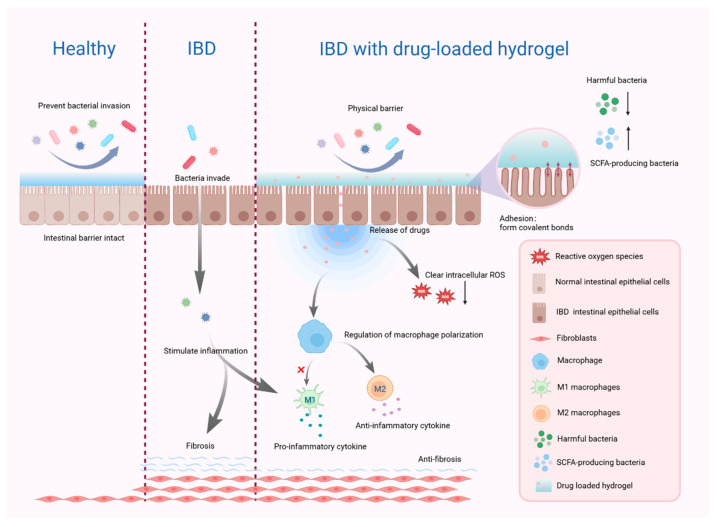
Drug-loaded hydrogels ameliorate IBD through multiple pathways. Created by Biorender.

**Table 1 ijms-26-02894-t001:** Advantages of hydrogels with different response mechanisms.

Response Mechanism	Advantages	Hydrogels	Mechanism	References
pH-sensitive	pH-sensitive hydrogels can release drugs at a specific PH.	HA-GE hydrogel	When pH = 6.8/7.4, the polyanionic nature facilitates drug release from the hydrogel via swelling and degradation of the hydrogel network.	[20]
		GA/SA hybrid hydrogel (GAS)	At lower pH, ionic interactions between hydrogen ions and deprotonated sodium alginate help maintain hydrogel stability.	[21]
Temperature-Sensitive	Thermosensitive hydrogels exhibiting excellent injectability and rapid, uncomplicated gelation properties, facilitating enema administration.	PDLLA-PEG-PDLLA	With the temperature change, sol-gel-sol conversion can be achieved.	[22]
		Thermosensitive injectable metabolic nanoregulator (TMNR)		[23]
Enzyme Sensitivity	Enzyme-sensitive hydrogels can achieve drug release in the colon microenvironment and improve targeting.	COS-CaP-QT	COS-CaP-QT microspheres are capable of colon-specific microenvironmental responses.	[24]
		AL/HA hydrogel	HA is susceptible to degradation by ROS and hyaluronidase in the intestines.	[25]
ROS Sensitive	ROS Sensitive hydrogels are selectively formed at inflammatory sites within the colon.	HASA	Sulfhydryl-substituted polymers can be oxidized to disulfide bonds by ROS to specifically form hydrogels.	[26]

**Table 2 ijms-26-02894-t002:** Application of drug-loaded hydrogel to alleviate inflammation and experimental subjects.

Effects	Hydrogels	Drugs	Subjects	References
Maintained the intestine barrier	PDLLA-PEG-PDLLA	Mesalazine	DSS-induced acute UC mouse models	[22]
	Inulin gel	Olsalazine	DSS-induced acute UC mouse models	[42]
			DSS-induced UC rats with delayed treatment	
			TNBS-induced CD mouse and rat models	
	PDLLA-PEG-PDLLA	Mn_3_O_4_ nanozyme	DSS-induced acute UC mouse models	[43]
	SAPH	-	TNBS-induced acute UC rat models	[44]
Regulate the immune system	HAMs	-	iBMDM cells DSS-induced acute UC mouse models	[45]
	SA, HA	Selenoprotein	iBMDM cells DSS-induced acute UC mouse models	[46]
	CS, SA	Emodin, curcumin	DSS-induced acute UC mouse models	[47]
Regulate gut microbiota	PEG hydrogel	UCB-MSCs	DSS-induced UC mouse models	[48]
	SA	Bioactive glass	DSS-induced acute UC mouse models	[49]
			DSS-induced chronic UC mouse models	
Clear intracellular ROS	CS-IGF-1C	hP-MSCs	TNBS-induced UC rat models	[50]
	SHARP	-	in vitro ROO·, ·OH, and O_2_^·−^ models	[51]
Adhesion	HA	Mesalazine	TNBS-induced UC rat models	[52]
Anti-fibrosis	SA	Bioactive glass	DSS-induced chronic UC mouse models	[49]
Relieve anxiety and depression	SP	Rh	DSS-induced chronic UC mouse models	[53]

**Table 3 ijms-26-02894-t003:** Advantages of different modes of administration.

Mode of Administration	Hydrogels	Advantages	References
Oral administration	SA	Hydrogels serve as carriers for oral medications, shielding them from breakdown caused by stomach acid and digestive enzymes, which enhances the drugs’ stability and efficiency.	[83]
	CS, SA		[84]
Rectal administration	AP	Incorporating drugs into hydrogels may extend their duration of action within the intestinal tract and improve their therapeutic effectiveness.	[85]
Endoscopic administration	Peptide-hydrogel	The gel’s characteristics can significantly enhance the duration of contact between the drug and the intestinal wall, thereby extending the therapeutic effect.	[86]
Subcutaneous administration	PEG-DPCA	Subcutaneous administration has a long time of action, slow release, and simple administration process, which can enhance patient adherence.	[87]

## Data Availability

Data sharing is not applicable to this article as no new data were created or analyzed in this study.

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
