# Peer review of "Recent Advances in the Application of Hydrogels as Drug Carriers in Inflammatory Bowel Disease: A Review"

_ijms, 2025, doi:10.3390/ijms26072894_

Round 1
Reviewer 1 Report
Comments and Suggestions for Authors
The paper is very interesting and innovative for the new suggestions for clinical practice in inflammatory bowel disease. I think that this work is fit for tthe publication in the journal but I suggest:
at page 1, line 2, I suggest to insert also Indeterminated Colitis
The Authors should insert a table with the charateristic of any hydrogel for any different therapeutical agents
Reviewer 2 Report
Comments and Suggestions for Authors
This review addresses the development and administration routes of hydrogels in the treatment of Inflammatory Bowel Disease (IBD). Although the study demonstrates relevance by discussing key research aimed at developing new therapies using hydrogels as drug delivery systems for IBD, I suggest that the author deeper into the discussion, particularly focusing on in vivo studies and/or preclinical trials that already explore the use of hydrogels in IBD. For greater relevance, I recommend a more thorough search and, if possible, the creation of a table showcasing studies that have already utilized hydrogels for this purpose. I believe that after improvements and corrections, this article could be a candidate for publication in IJMS.
Line 22-23 – I suggest rewriting the phrase "primarily manifests through symptoms including weight loss, rec- 22 tal bleeding, and diarrhea" to "primarily manifests through symptoms including weight loss, rectal bleeding, and diarrhea."
Line 24-25 – I suggest that the authors place the references at the end of the citation. Please [2,3].
Line 36 – Please elaborate for the reader on the following citation: “For instance, biologic therapies are associated with an elevated risk of serious infection.” What leads to this risk of infection?
Line 39 – I suggest removing the comma in the citation “osteoporosis, and increased infection susceptibility” and rewriting it as “osteoporosis and/or increased infection susceptibility.”
Line 42-43 – I suggest the authors replace the word “ameliorate” with “improve” in the sentence.
Line 47-48 – I suggest the authors rewrite the sentence to “Hydrogels are three-dimensional networks composed of hydrophilic polymers, which have the ability to absorb and retain large amounts of water.”
Line 108-110 – I suggest the authors verify this sentence, as there is a duplication of the citation.
Line 114 – I suggest the authors verify if the citation for this paragraph is correct, as well as if the citation "Guo et al." is updated and accurate. Please italicize the word “in vivo”
Line 131 – Please correct the phrase “and At room temperature” by removing the uppercase word.
Line 198 – I suggest the authors cite the figure, whether it is original, adapted, or developed using software. I also recommend that the authors check the numbering of the figures, as there are two Figure 1s in the article.
Line 203 – Please include the citation for this sentence.
Line 214-216 – Please correct the necessary words to italics: “in vitro,” “S. aureus,” and “E. coli.”
Line 433-439 – Please insert the corresponding citation for this sentence.
Round 2
Reviewer 2 Report
Comments and Suggestions for Authors
Dear authors,
Thank you for making the proposed changes. I believe that after this process the article has potential for publication. I would just like to point out to the authors that they should reevaluate whether Figures 1 and 2 require citation of the software that was used to develop these images (e.g. "created with Biorender.com"); due to licensing issues.